# Yeast Extract Affecting the Transformation of Biogenic Tooeleite and Its Stability

**Qingzhu Li** [1,2,3], **Qianwen Liu** [1], **Xi Wang** [1], **Qi Liao** [1,2,3], **Hui Liu** [1,2,3] **and Qingwei Wang** [1,2,3,*]

1    School of Metallurgy and Environment, Central South University, Changsha 410083, China;
     qingzhuli@csu.edu.cn (Q.L.); liu_qw@csu.edu.cn (Q.L.); csu13187075865@163.com (X.W.);
     liaoqi@csu.edu.cn (Q.L.); leolau@csu.edu.cn (H.L.)
2    Chinese National Engineering Research Center for Control and Treatment of Heavy Metal Pollution,
     Changsha 410083, China
3    Water Pollution Control Technology Key Lab of Hunan Province, Changsha 410083, China
*    Correspondence: qw_wang@csu.edu.cn; Tel.: +86-731-89873970

**Abstract:** Highly toxic As(III) is the main form of As in wastewater. The retention of As by tooeleite has gradually attracted attention in recent years due to its great potential for the direct removal of As(III). The existence of natural As-bearing minerals is closely related to microorganisms and organic matters. In this study, yeast extract was found to enhance the stability of biogenic tooeleite by *Acidithiobacillus ferrooxidans* (*A. ferrooxidans*). The effects of pH, Fe/As and yeast extract concentration were systematically studied, and the toxicity characteristic leaching procedure (TCLP) was conducted to evaluate the short-term stability of tooeleite. The mineral synthesized in the presence of yeast extract showed that the As leaching concentration decreased from 13.78 mg/L to 7.23 mg/L and the stability increased by more than 40%. In addition, various characteristics confirmed that the precursor was changed from amorphous schwertmannite to basic ferric sulfate in the presence of yeast extract, and then transformed to relatively purer tooeleite with less hollow structure and excellent dispersion, which is favorable for the stability of tooeleite. This result indicated that yeast extract resulted in the formation of different precursors and thus affected the transformation and stability of tooeleite.

**Keywords:** tooeleite; biogenic; stability; transformation; yeast extract

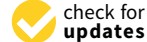



## 1. Introduction

Highly toxic and soluble arsenite (As(III)) is a predominant arsenic species in acid mine drainage (AMD) and its appropriate immobilization is important [1–3]. Moreover, various secondary Fe(III) minerals are found extensively in AMD, such as schwertmannite, jarosite, and goethite, which may play a key role in the removal of As [4,5]. Although schwertmannite can adsorb up to 70 mg/g As(V), higher As concentrations inhibit its crystallization [6]. Savage et al. [7] formed jarosite to remove As(V) by incorporating As into the mineral structure by substitution of the S site. Nonetheless, it is not stable under any reservoir conditions and As could be released into the environment. Since many techniques used for the removal of As(III) from aqueous solutions require a pretreatment process for chemical oxidation, there is an urgent need for direct treatment of As(III) and safe stability.

Tooeleite ($Fe_6(AsO_3)_4SO_4(OH)_4 \cdot 4H_2O$), the only ferric arsenite sulfate mineral currently known, has a high As content (20–25 wt.%) [8]. It has attracted mineral and metallurgical researchers for its potential significance in As(III) treatment [9]. In previous studies, several chemical and biological methods to form tooeleite have been reported, but most kinds of synthetic tooeleite were not involved in stability research [10–12]. However, Raghav et al. [8] studied the toxicity of tooeleite synthesized at 60 °C and pH 2.6 with or without silica coating. Although the silica-coating significantly reduced the As leaching of tooeleite, both the pure and silica-amended tooeleite still showed much higher As leaching

concentrations than 5 mg/L [13]. Furthermore, the stability of tooeleite could be enhanced through a hydrothermal method considering the influence of critical factors, such as purity and crystallinity [14,15]. Currently, such methods for improving mineral stability are not suitable and economical.

Fe(II)-oxidizing microbes play a significant role in As attenuation through biomineralization of secondary Fe(III) minerals [4]. Biogenic tooeleite is thought to be more stable than chemogenic tooeleite due to the existence of a large quantity of natural minerals related to bacteria [11]. *Acidithiobacillus ferrooxidans* is a typical acidophilic chemolithoautotrophic bacteria [16,17]. Egal [5] found that *A. ferrooxidans* can oxidize Fe(II) to promote As(III)-rich precipitates, composed of As(III)-Fe(III) amorphous phase, schwertmannite, jarosite and tooeleite. Cesbron and Williams [18] first reported that tooeleite occurred in oxidized mine tailings and that primary ores were rich in pyrite and arsenopyrite. It has been reported that arsenopyrite was preferentially dissolved and tooeleite precipitated during bioleaching [19]. In addition, bacteria are effectively used in metal sulfide leaching, and extracellular polymeric substances (EPS) seem to play a pivotal role in dissolution by creating a micro reaction space [20,21]. The organic region could also be the place where the secondary mineral was formed.

Indeed, organic matter (OM) has a significant impact on regulating and inducing the formation of minerals. Iron minerals show a high reactivity towards dissolved organic matter [22–24]. Moreover, the oriented attachment of organic matter may lead to the specific morphology of mineral particles [25,26]. The stability of schwertmannite can be improved by elevated concentrations of natural organic matter [27,28]. Interestingly, Okibe [29] found that organic matter (0.01–0.05% (*w/v*)) can increase biomass of thermo-acidophilic iron-oxidizing archaeon (*Ac. Brierleyi*) as energy source and provide an environment which promotes the As(III) oxidation reaction (e.g., the EPS (Extracellular Polymeric Substances) region around the cell surface. So it can improve not only scorodite crystallinity, but also As immobilization efficiency (up to 100%). To date, information concerned with the effect of organic matter on tooeleite transformation and stability has not been found in the literature.

Therefore, laboratory experiments were conducted to investigate the transformation and stability of biogenic tooeleite by adding *Acidithiobacillus ferrooxidans* ATCC 23270. Then, the influence of yeast extract on the stability and transformation behavior of biogenic tooeleite was discussed. The results would improve the understanding of the role of liquid phase organic matter on the formation of natural tooeleite.

## 2. Materials and Methods

### 2.1. Microorganism

*Acidithiobacillus ferrooxidans* ATCC 23270 (*A. ferrooxidans*) was cultivated in 9K medium with 44.78 g/L $FeSO_4 \cdot 7H_2O$ and 1.0 g/L As(III) solution at 30 °C on a rotary shaker (170 rpm). The basal salts in 9K medium consisted of 0.5 g/L $(NH_4)_2SO_4$, 0.5 g/L $MgSO_4 \cdot 7H_2O$, 0.5 g/L $KH_2PO_4$, and 0.1 g/L KCl. The solution was adjusted to pH 2.0 with $H_2SO_4$ and sterilized at 120 °C for 15–20 min before inoculation. The initial concentration of *A. ferrooxidans* after inoculation was $10^5$ cells/mL counted by microscopic counting method. The bacteria were cultivated until Fe(II) oxidation was completed (i.e., bacteria entered into the stationary stage, ~3 days).

### 2.2. Biotooeleite Crystallization Experiment

The bacterial suspension (10% (*v/v*)) was inoculated into 300 mL flasks after being filtered by Whatman filter paper to remove precipitates. The 300 mL flasks contained 100 mL of pre-sterilized medium solution containing As(III) (13.3 mM), Fe(II), and yeast extract. An initial As(III) concentration of 13.3 mM was chosen because such high enrichment of up to 1 g As(III)/L has been reported at mine drainage sites [30–33]. The mixed solution was stirred at 30 °C on a rotary shaker (170 rpm) for 4 days. The influence of pH from 1.0 to 6.0, Fe/As ratio from 1 to 10 was systematically studied to verify optimum condition for arsenic removal. Then, yeast extract was added at such optimum condition



with concentration from 0.00% to 0.10% (wt/vol). Yeast extract is a complex mixture of organic compounds obtained from hydrolyzed or autolyzed yeast cells. The different sampling time was designed to further verify the role of yeast extract at 3, 6, 12, and 24 h. The initial solution with or without 0.10% yeast extract both contained 13.3 mM As(III) and 79.8 mM Fe(II), at pH of 2.0. All experiments were performed in duplicate.

*2.3. Analytical Methods*

Liquid samples were filtered to determine the concentrations of total As and Fe by inductively coupled plasma-optical emission spectrometer (ICP-OES, ICAP 7000, Thermo Fisher Scientific CN, USA). Concentration of dissolved Fe(II) was measured by potassium dichromate titration [34]. The pH value was measured by a PHSJ-4F equipped with a pH sensor. The filter residues were washed with deionized water for three times, dried at 60 °C under vacuum and then ground for further analyses.

X-ray powder diffraction (XRD, TTR-III, Rigaku, Japan) was scanned from 5° to 70° with a stepped increment of 0.02° and 0.12 s counting time. Data were collected using Rigaku D/Max-RB diffractometer with Cu-Kα radiation (λ = 0.15406 nm, 40 kV, 250 mA). The valence of the arsenic was identified with X-ray photoelectron spectroscopy (XPS, Thermo Fisher Scientific K-Alpha 1063, USA). Microscopic observations were conducted with scanning electron microscopy (SEM, JSM-IT300LA, JEOL, Japan). The functional groups of solid samples were distinguished through Fourier transformed infrared spectroscopy (FTIR, NicoletIS10, Thermo Electron Corporation, USA) at a resolution of 4 cm$^{-1}$. To determine the change of element content in solids, the solids were digested 6M HCl for 12 h, the concentrations of total As Fe and S were then detected by ICP-OES (ICAP 7000).

The particle size measurements were made with laser diffraction particle size analyzer (LPSA, LS-pop (6), OMEC, China) and the BET specific surface area (SSA) was determined by nitrogen adsorption (model Nova 1000, Quantachrome, USA).

*2.4. Toxicity Characteristic Leaching Procedure (TCLP) Test*

The TCLP test was conducted by following EPA method 13111, using an extraction fluid of pH 4.93 ± 0.05. The extraction fluid consists of 5.7 mL of acetic acid (CH$_3$CH$_2$COOH) and 64.3 mL of 1 M NaOH, and has a final volume of 1000 mL. The solid samples were mixed with the extraction fluid in a solid-to-liquid ratio of 1:20 in 50 mL centrifuge tubes, and then continuously shaken on a 30-rpm tumble shaker at 25 °C for 18 ± 2 h. Suspensions were finally filtered using 0.45 μm membrane filters. The concentrations of total As and Fe in lixivium were analyzed by ICP-OES.

## 3. Results and Discussion
### 3.1. Biogenic Tooeleite Formation

To ensure the efficient removal of arsenic before adding yeast extract and highlight the potential of yeast extract for As stabilization, the optimum condition of biogenic tooeleite need to be determined firstly.

#### 3.1.1. Effect of the Initial pH

pH is one of the most important factors for the removal of As and the formation of biogenic tooeleite because it influences iron species and bacterial growth. Furthermore, the effect of the initial pH on the precipitation process was studied in the range of 1.0–6.0.

The chemical equation of Fe(II) oxidation and Fe(III) hydrolysis can be expressed as follows:

$$Fe^{2+} + O_2 + H^+ \xrightarrow{A.f} Fe^{3+} + H_2O \tag{1}$$

$$Fe^{3+} + H_2O \rightarrow FeOH^{2+} + H^+ \tag{2}$$

Iron oxidation consumes H$^+$ and increases pH while the hydrolysis of Fe(III) counteracts the increase in pH of the solution. Figure 1a shows that the concentrations of Fe(II) all decreased and Fe(II) was completely oxidized on the second day at pH = 2.0~6.0. The

abiotic oxidation of Fe(II) was very slow at pH < 3, but Fe(II) oxidation could be accelerated by *A. ferrooxidans* [35]. Fe(II) oxidation rate was relatively slow during 4 d at initial pH = 1.0 as bacterial activity was reduced under such low-pH conditions. In contrast, the Fe(II) oxidation rate was significantly higher when the initial pH was 6.0, which was the dominant oxidation of Fe(II) by oxygen in the air [36]. At the same time, the pH decreased to 3.5 after 1 d indicating that the hydrolysis of iron was far stronger than the oxidation of Fe(II) (Figure 1b). The pH of the system remained stable at approximately 2.0 after 2 d when the initial pH 2.0~4.0. Hence, it can be concluded that a suitable initial pH range (2.0~4.0) could slow down the local pH mutation, providing a suitable micro-environment for the normal growth of bacteria and the stability of system supersaturation [37].

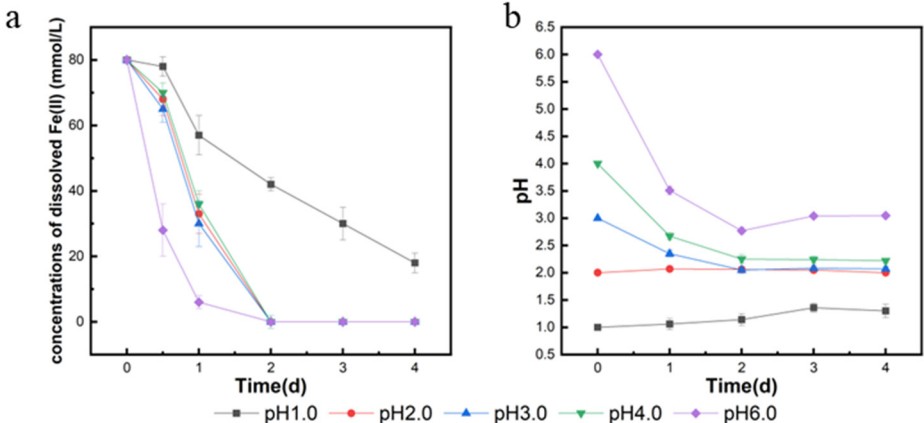

**Figure 1.** Concentrations of dissolved Fe(II) (**a**) and pH change (**b**) in the biogenic tooeleite process under different initial pH conditions.

The removal efficiency of As and the concentration of As leaching are illustrated in Figure 2a. The As removal efficiency apparently increased with increasing pH from 1.0 to 1.8 while it decreased slightly at pH 2–6. The XRD patterns (Figure 2b) confirmed that tooeleite was the only detectable mineral consistent with PDF 44–1468. This proved the formation of tooeleite in the pH range of 1.8–4.0, which is wider than that in a previous report (2.0–3.5) [9]. The analysis of XRD patterns corresponding to pH revealed that acidic conditions lead to better crystallization and the optimal crystallized tooeleite was obtained at pH = 2.0. Meanwhile, the concentration of As leaching was as low as approximately 13 mg/L. The above observations showed that a suitable initial pH range provided suitable ore-forming conditions for tooeleite. Therefore, pH 2.0 was chosen for capably treating acid waste-water and synthesizing relatively stable tooeleite.

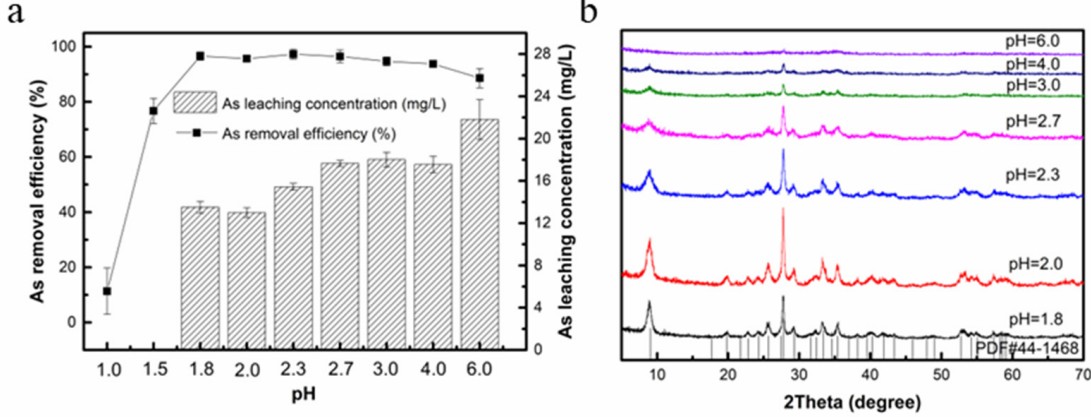

**Figure 2.** As removal efficiency, As leaching concentration (**a**) and XRD patterns of the precipitates (**b**) obtained at different initial pH values.

### 3.1.2. Effect of Fe/As

Fe(II) is the energy source needed for the growth of bacteria, and the removal of As has a heavy dependence on iron ions. Iron ions could be concentrated when *A. ferrooxidans* grows [38,39], and biogenic tooeleite formation may be triggered in the EPS region where metal species can be more susceptible to chemical reactions than in bulk solutions [40]. Therefore, keeping the As concentration constant, the effect of the ratio of Fe to As on the precipitation of tooeleite was studied in Fe/As ranging from 1 to 10 at pH 2.0.

As shown in Figure 3b, tooeleite was the only mineral detectable at different Fe/As ratios. The lowest concentration of As leaching was shown at Fe/As = 2 and the precipitation crystallinity was relatively the best (Figure 3). It was consistent with a previous report, that is, the chemical tooeleite crystalline degree can be improved with an increase in Fe/As from 0.8 to 2 [10]. However, the removal efficiency of As stayed below 80% under the above conditions, and it reached over 95% under Fe/As > 6 (Figure 3a). So it was suggested that the higher Fe/As benefited for bacterial growth and efficiently As removal. The above observations showed that excess Fe(II) (closer to (Fe[II])ini/(As[III])ini = 6) was found beneficial to improve the final As removal (up to 98–99%), but detrimental to decrease the As leaching concentration.

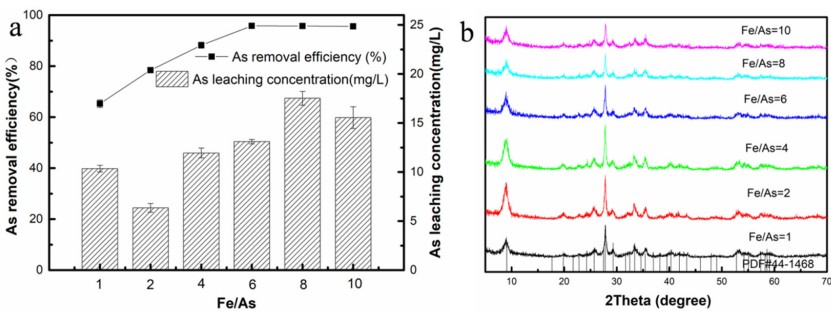

**Figure 3.** As removal efficiency, As leaching concentration (**a**) and XRD patterns of the precipitates (**b**) obtained under different Fe/As ratios.

### 3.1.3. XPS Analysis of the Formed Precipitate

To further confirm the valence of As in precipitates during the oxidation of Fe(II) by *A. ferrooxidans.* The XPS spectrum for the sample obtained at pH 2.0 and Fe/As = 6 is shown in Figure 4. An obvious peak is observed with binding energy of 44.03 (eV) which indicates that the only valence of As in the precipitate obtained under optimal conditions is +3 [41,42]. It shows certain potential for the direct removal of As(III) by the formation of biogenic tooeleite. The slightly higher As leaching concentration of tooeleite may substantially restrict its application. Thus, the stability of tooeleite was improved by adding yeast extract in the following experiment.

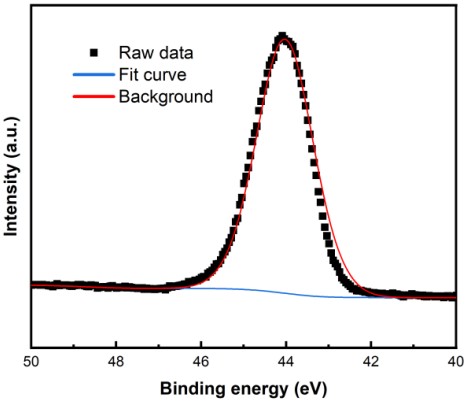

**Figure 4.** XPS analysis of the precipitate obtained under optimum conditions.

### 3.2. The Effect of Yeast Extract on the Stability of Tooeleite

The presence of yeast extract in aquatic systems has the potential to affect the stability of biogenic mineral [29]. Therefore, the effects of 0.00, 0.02, 0.06, and 0.10% (*w/v*) yeast extract were investigated in the present study (Fe$^{2+}$ oxidation will be hindered as yeast extract (>0.10%) would inactivate bacteria). Furthermore, 0.10% yeast extract had no effect on bacterial growth (Figure S2). Additionally, in experiments conducted in the absence of bacteria, Fe(II) and As(III) concentrations were nearly constant [15]. Figure 5b shows the XRD pattern of the precipitate obtained under different concentrations of yeast extract. Tooeleite was the only mineral detectable, whose characteristic peaks were consistent with PDF 44–1468 [18]. Figure 5a shows that the As removal efficiency of the system remained above 95% with increasing yeast extract concentration. The As leaching concentration gradually decreased from 13.8 mg/L to 7.2 mg/L and the stability improved by approximately 40%. The main aspects that remained to be explained in the present study are the stabilization mechanism and the role of yeast extract in the reaction process.

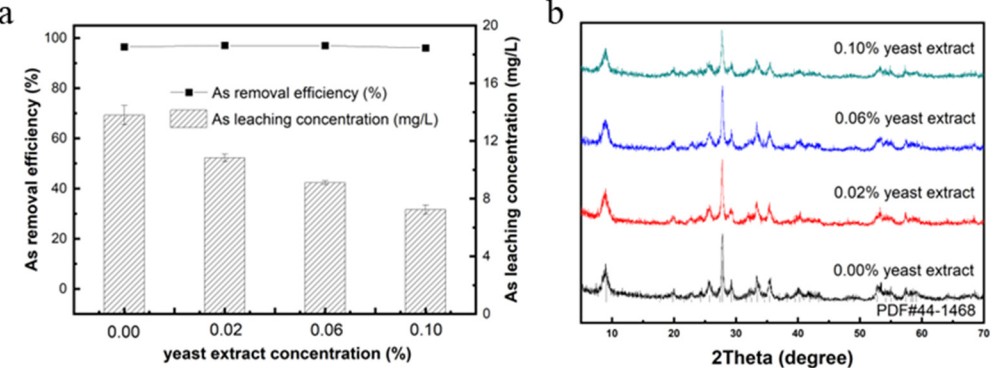

**Figure 5.** As removal efficiency, As leaching concentration (**a**) and XRD patterns of the precipitates (**b**) under different yeast extract concentrations.

### 3.3. The Role of Yeast Extract during the Formation of Tooeleite and As Stabilization Mechanism

3.3.1. Effect of Yeast Extract on Tooeleite Transformation

The XRD patterns of the precipitate obtained at different times without (a) or with (b) yeast extract are shown in Figure 6. This result revealed that the precipitation formed in the initial stage of the reaction was an amorphous phase. Figure 6 shows that typical tooeleite peaks were detected at 6 h in absence of yeast extract, but a peak obtained at 6 h exhibits a weak band at 27° in the presence of yeast extract. The latter XRD patterns display intensity of tooeleite peaks increased over time, and no other crystals occurred during the reaction. As observed above, it is clear that the incorporation of yeast extract delays tooeleite formation.

Figure 7 shows the SEM images of tooeleite particles prepared without or with yeast extract. It can be seen from Figure 7(a-1,a-5,a-10) that the form of this precipitate is the flower structure of platy particles aggregating together, which is consistent with previous research [43]. However, the SEM results of tooeleite synthesized in the presence of yeast extract showed that the platy structure of the final product was thicker and clearer (Figure 7(b-5,b-10)). It has been reported that different transformation pathways affect the final mineral morphology [44,45]. Furthermore, yeast extract may play an important role in the mineral transformation process so that tooeleite particles with such a specific morphology could be formed.

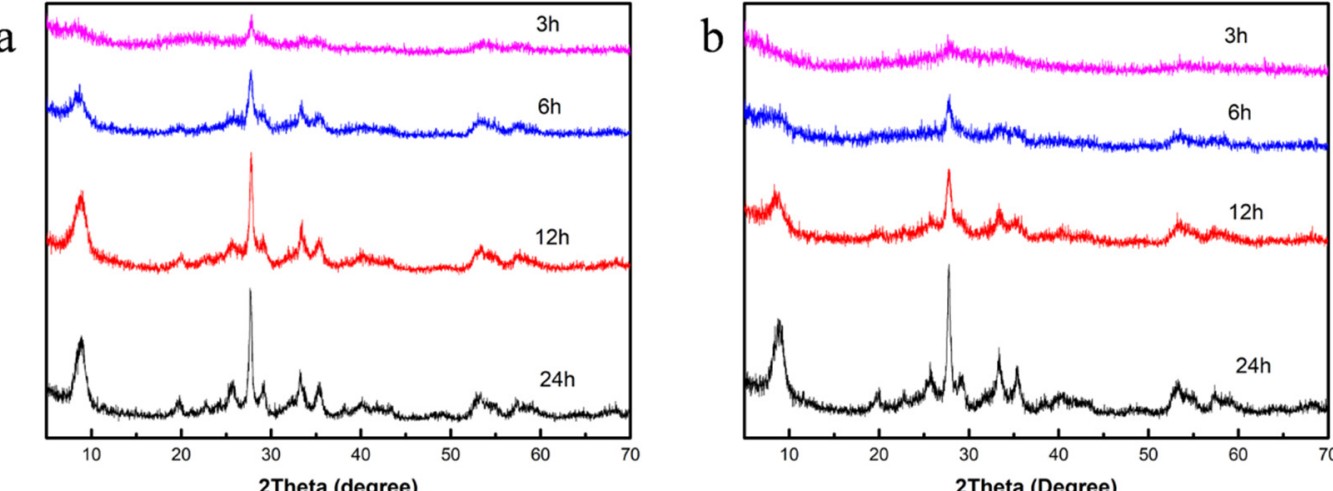

**Figure 6.** XRD patterns of the precipitates obtained under 0.00% (**a**) and 0.10% (**b**) yeast extract concentrations at different reaction times.

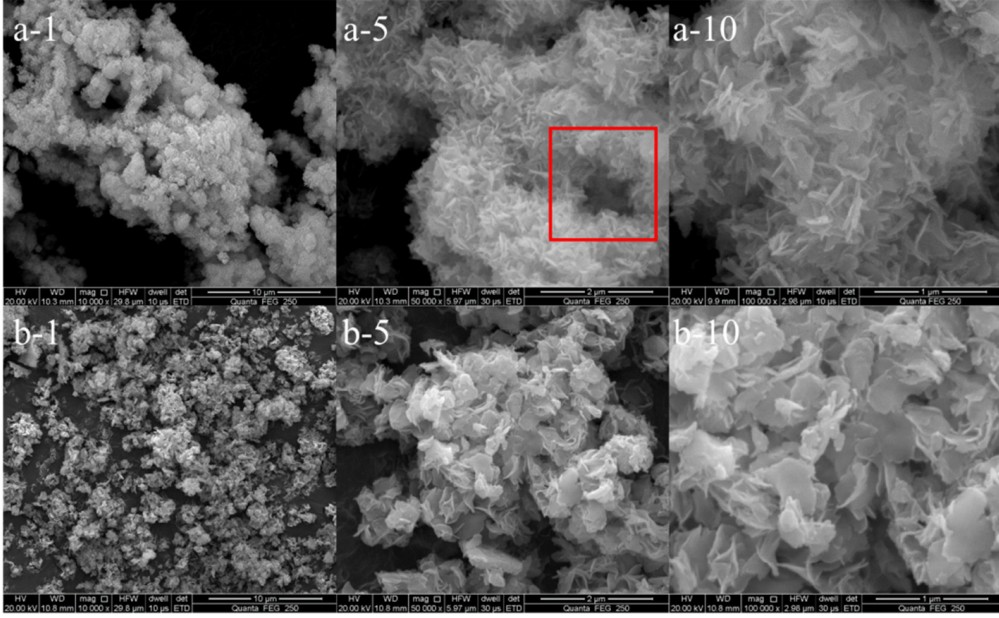

**Figure 7.** SEM images of precipitates under optimum conditions (hollow structure marked by a red box), (a-1), (a-5), and (a-10) at magnifications of 10,000, 50,000, and 100,000 without yeast extract; (b-1), (b-5), and (b-10) at corresponding magnifications with yeast extract.

The FTIR spectra of samples collected at different times without (Figure 8a,b) and with (Figure 8c,d) yeast extract revealed changes in the functional groups of precipitates during mineral transformation. As shown in Figure 8a,c, the FTIR spectra of precipitate shows gradually enhanced absorption peaks at 750~950 cm$^{-1}$, 686~692 cm$^{-1}$, and 511~513 cm$^{-1}$ which belong to the stretching vibration peak of As-O [46], Fe-O, and Fe-O-As [47]. The above bands ascribed to tooeleite and their intensities increased gradually during 24 h [48].

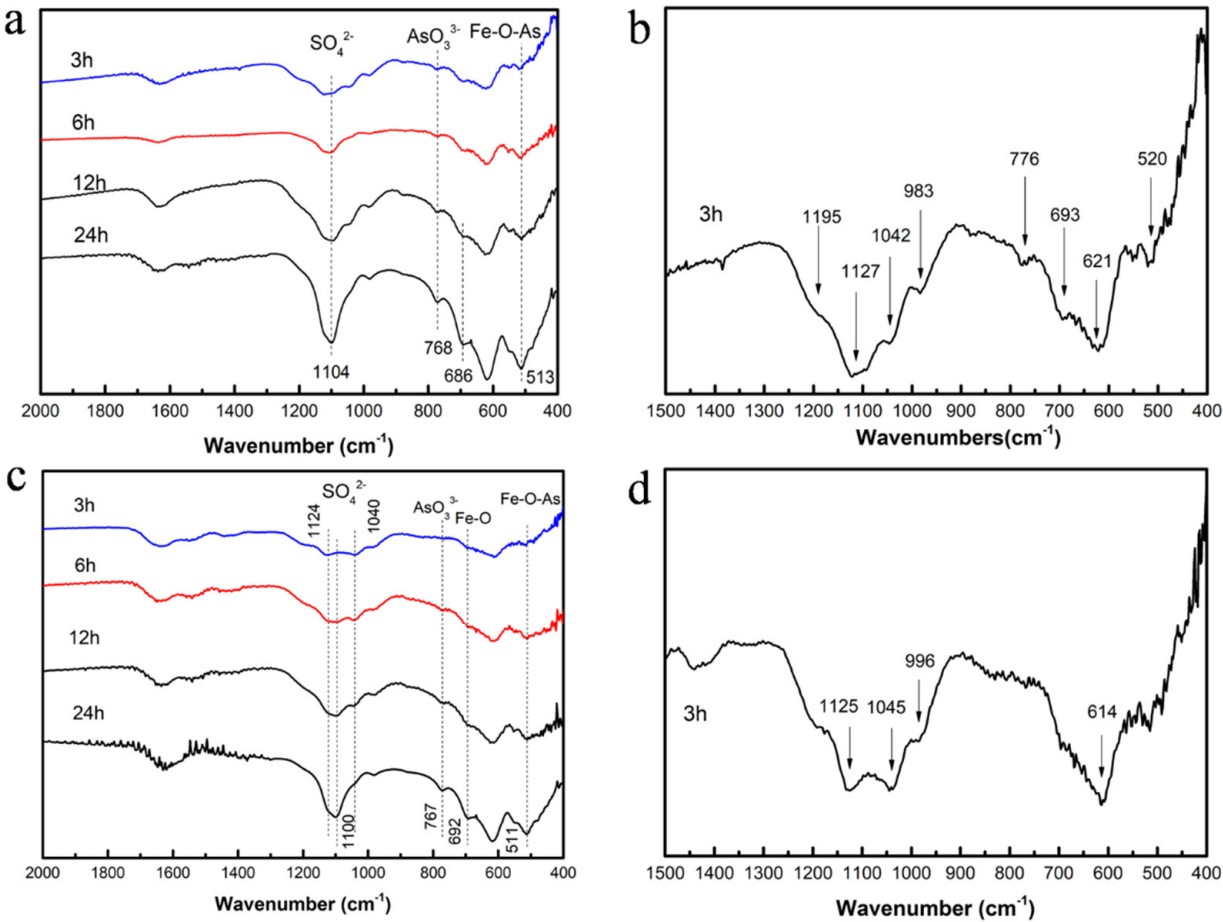

**Figure 8.** FTIR analysis of precipitates at different reaction time: (**a**,**b**)—without yeast extract; (**c**,**d**)—with yeast extract.

Furthermore, the XRD patterns of the precipitates obtained at 3 h with yeast extract does not look much different than that without yeast extract, both precipitates were amorphous phase (Figure 6). So it was important to verify the functional groups of precipitates formed at 3 h. In Figure 8b, a sharp band at 1127 cm$^{-1}$ and two shoulders at 1042 cm$^{-1}$ and 1195 cm$^{-1}$ can be attributed to the $v_3$ triply degenerate antisymmetric stretching vibration of $SO_4^{2-}$ in schwertmannite [49]. The removal of As by schwertmannite takes place only at pH values from 2 to 7 [50], which further proved that the precursor was schwertmannite adsorbing As. In Figure 8d, two small bands (1124 cm$^{-1}$ and 1040 cm$^{-1}$) appeared which corresponded to the substances produced by the interaction of $SO_4^{2-}$, $Fe^{3+}$, and $OH^-$ (1058 and 1112 cm$^{-1}$) [51]. It was demonstrated that the amorphous material was most likely basic ferric sulfate ($MFe_x(SO_4)_y(OH)_z$, $M^+ = K^+$, $Na^+$, $NH_4^+$) [51,52].

Although, in the system without yeast extract, the bands at 1195 cm$^{-1}$, 1127 cm$^{-1}$, and 1042 cm$^{-1}$ gradually disappeared, the band at 1104 cm$^{-1}$ remained over time (Figure 8a). Additionally, as shown in Figure 8c, two bands (1124 cm$^{-1}$ and 1040 cm$^{-1}$) were converted into another band (1100 cm$^{-1}$), these bands all belong to $SO_4^{2-}$. The results indicated that different phases (schwertmannite and basic ferric sulfate) of early precipitates formed in both systems during the transformation process.

Compared to the reaction without yeast extract, the higher organic content in precipitates obtained with yeast extract at 3 h (Figure S1 and Table S1) indicates that yeast extract plays an important role during the initial stage of reaction. As EPS are a complex high-molecular-weight mixture of polymers excreted by microorganisms, produced from cell lysis and adsorbed organic matter from the external environment. It can be suggested that yeast extract provides cells with a suitable EPS reaction region where ions can be con-

centrated by complexation during bacteria growth [29,44,45]. The nucleation of inorganic metal compounds was affected by biomacromolecules as organic templates [53–56]. As the extension of reaction time, arsenic content in solids obtained with YE increased, which were not shown in the reaction without YE (Table S2). Above all, amorphous schwertmannite adsorbing As was formed and then transformed to tooeleite in absence of yeast extract, whereas tooeleite was transformed from basic ferric sulfate in the presence of yeast extract. The results led us to the conclusion that yeast extract provided the bacteria with a suitable EPS reaction region where iron ions can be concentrated and basic ferric sulfate nucleated as the precursor. The different precursors affected the mineral transformation process and then affected the morphology of tooeleite.

### 3.3.2. Effect of Yeast Extract on Tooeleite Stability

The As solubility of the As-bearing precipitates is one of the important parameters that must be assessed. The concentration of As leached from precipitates decreased from 13.8 mg/L to 7.2 mg/L when 0.10% yeast extract was introduced into the system. The average particle size and specific surface area of the former were 10.84 μm and 67 m$^2$/g, but those of the latter were 5.12 μm and 71 m$^2$/g, respectively (Figure 9). Figure 7(a-1,b-1) shows that the particles prepared in absence of yeast extract were obviously agglomerative, which was consistent with the results of a larger particle size. The results showed that the specific surface area of biogenic tooeleite changed slightly, while the average particle size of tooeleite prepared in the absence of yeast extract was nearly two-times larger than that of tooeleite prepared in the presence of yeast extract. These results may be due to the different morphologies of the particles. With more hollow structures appearing, the surface roughness of particles increased in tooeleite without yeast extract. Therefore, the rougher surface of the tooeleite without yeast extract increased the specific surface area similar to that of the tooeleite with yeast extract, although the particle size varied greatly. In addition, it has been reported that the hollow structure of particles leads to a relatively higher As leaching concentration [14,57]. This is the reason that both external diffusion and internal diffusion exist in the leaching reaction of the precipitates with hollow structures. Here, it can be suggested that yeast extract is related to the change in the morphology of the precipitates, and the lower As leaching concentration is attributed to the distinct mineral morphology with less hollow structure.

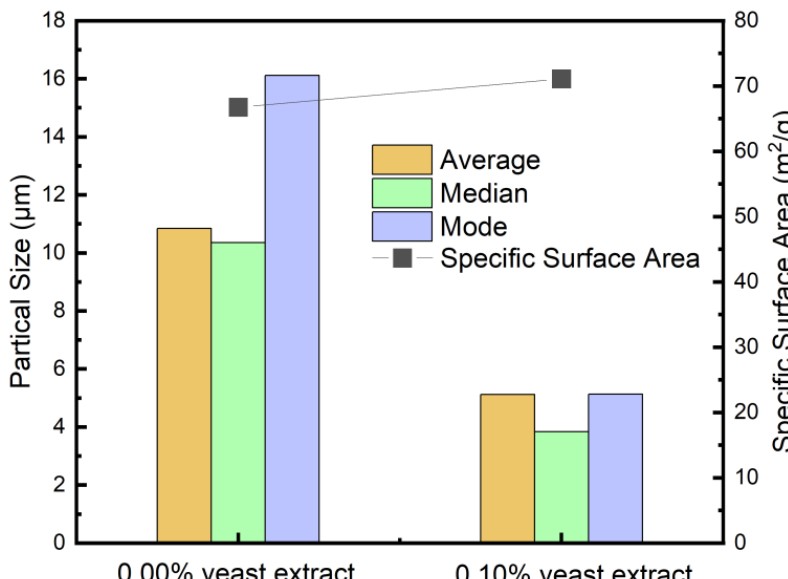

**Figure 9.** Particle size and specific surface area of biogenic tooeleite obtained with or without yeast extract (YE).

The elemental composition of tooeleite is shown in Table 1. When the yeast extract was added or not, the Fe/As of the products became different. In the system without yeast extract, the Fe/As ratio of the precipitate was 2.17, which is obviously above the theoretical value of the tooeleite molecular formula: 1.50. This illustrated that there was amorphous iron oxide in the final product except tooeleite. In addition, the molar Fe/As ratio of the resulting product in the system with yeast extract was 1.60, which was much closer to 1.50. Based on these results, yeast extract could contribute to purer tooeleite formation, which is favorable for the stability of tooeleite.

**Table 1.** ICP analysis results of precipitates obtained with or without yeast extract (YE).

| YE Concentration (*w/v*) | As (*w/w*) | Fe (*w/w*) | S (*w/w*) | Fe/As (Molar Ratio) |
|---|---|---|---|---|
| 0.00% YE | 16.24% | 26.26% | 3.79% | 2.17 |
| 0.10% YE | 21.71% | 25.97% | 3.90% | 1.60 |

*3.4. Environmental Implications*

Elevated As concentrations are usually found in AMD so it is important to find a suitable method to control it. Tooeleite ($Fe_6(AsO_3)_4SO_4(OH)_4 \cdot 4H_2O$), as a naturally formed As-mineral with a high As content of 20–25 wt%, can be considered a temporary scavenger for As presenting in acidic conditions and plays an important role in As geochemical cycling. Among the current treatment processes for As(III) control, immobilization of As(III) by tooeleite may be considered a promising technology. However, few studies focus on the stability of tooeleite, and this is vital for further application. In previous literature, the leached As can be below 5 mg/L at high initial As concentration (up to 3–4.5 g/L) by coating tooleite with silica or oxidizing Fe(II) slowly [8,15]. This is not inapplicable to AMD system. In our research, yeast extract can improve the stability of solids without affecting As removal efficiency under initial As concentration of 13.3 mM.

It was suggested that organic matter plays an important role in regulating and inducing the formation of minerals. There is little available information on organic matter influencing the transformation and stability of tooeleite. As shown in the present study, biogenic tooeleite formation by *A. ferrooxidans* led to over 95% As removal efficiency, and the incorporation of yeast extract into tooeleite improved its stability by 40%. A schematic diagram of yeast extract affecting the formation of biogenic tooeleite is shown in Figure 10. The As retention capacity of tooeleite may be impacted by the transformation process. Different precursors obtained in the presence of yeast extract will affect mineral transformation, and then relatively purer tooeleite with a specific morphology will be formed to enhance its stability. This study demonstrated that yeast extract could enhance tooeleite stability with better removal and fixation ability of As(III) from high-concentration As acid drainage. However, future studies must be carried out to substantiate details of the treatment process.

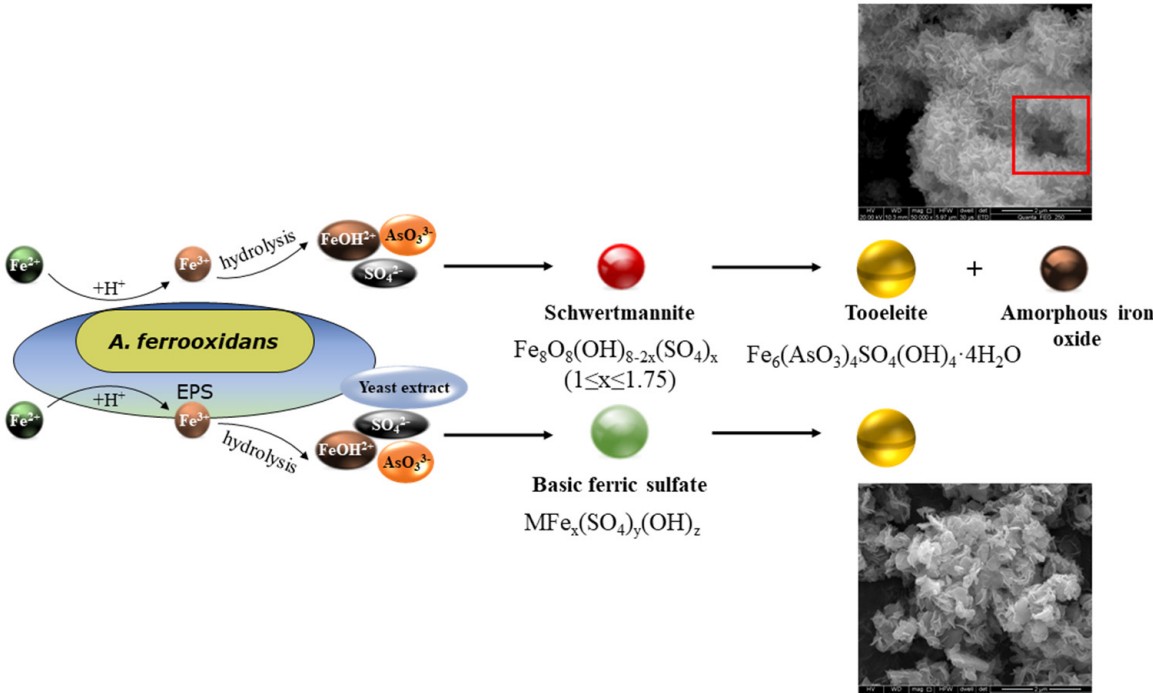

**Figure 10.** Schematic diagram of yeast extract affecting the formation of biogenic tooeleite.

### 4. Conclusions

Here, the influence of organic matter during the reaction on the mineral transformation and the stability of tooeleite were investigated. The optimum conditions for biogenic tooeleite formation at room temperature were pH 2.0 and Fe/As = 6. It was found that the concentration of As leached from the precipitate diminished to 7.2 mg/L as the yeast extract content increased to 0.10%.

Furthermore, the mineral transformation process was revealed. At the beginning of the reaction, amorphous schwertmannite was formed and then transformed to tooeleite in the system without yeast extract. However, yeast extract induced the nucleation of basic ferric sulfate as the precursor, and then the precursor transformed to tooeleite. Different transformation processes would affect the final mineral morphology. The platy structure of tooeleite obtained with yeast extract is thicker and clearer. In addition, yeast extract led to relatively purer tooeleite with a less hollow structure and excellent dispersion, which implied relatively lower As leaching concentration and enhanced stability of tooeleite. In conclusion, yeast extract affected mineral transformation and enhanced As(III) stability because it changed nucleation and crystallization during the formation of tooeleite.

**Supplementary Materials:** The following supporting information can be downloaded at: https://www.mdpi.com/article/10.3390/app12073290/s1, Figure S1: SEM images and EDS analysis of precipitates obtained without (a) or with (b) yeast extract at 3 h; Figure S2: Dynamic changes of OD600 under 0.00% and 0.10% yeast extract; Table S1: EDS analysis results of precipitates obtained with and without yeast extract (YE) at 3 h; Table S2: ICP analysis results of precipitates obtained with (a) and without (b) yeast extract (YE) at different reaction time.

**Author Contributions:** Conceptualization, Q.L. (Qingzhu Li); methodology, Q.L. (Qianwen Liu); formal analysis, Q.L. (Qianwen Liu) and Q.L. (Qingzhu Li); software, X.W.; validation, Q.L. (Qi Liao) and H.L.; investigation, Q.W. All authors have read and agreed to the published version of the manuscript.

**Funding:** This research was supported by the National Natural Science Foundation of China (51974379), the Project of National Science Fund for Excellent Young Scholars of China (52022111), Huxiang Youth Talent Support Program (2020RC3012).

**Institutional Review Board Statement:** Not applicable.

**Informed Consent Statement:** Not applicable.

**Data Availability Statement:** Data is contained within the article.

**Acknowledgments:** We want to acknowledge the National Natural Science Foundation of China (51974379), the Project of National Science Fund for Excellent Young Scholars of China (52022111), Huxiang Youth Talent Support Program (2020RC3012).

**Conflicts of Interest:** The authors declare no conflict of interest.

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
