# Peer review of "Yeast Extract Affecting the Transformation of Biogenic Tooeleite and Its Stability"

_applsci, doi:10.3390/app12073290_

Round 1

Reviewer 1 Report

This is an interesting, detailed study in environmental geomicrobiology. The approach is classic, with little novelty, but research related to remediation of As contamination in water is in constant need. Therefore, tests on new combined systems are needed. The manuscript requires content-specific corrections and additions as well as editorial adjustments.

Specific comments:

  1. Section: 2.2. Biotooeleite crystallization experiment

Please give an explanation as to what the rationale was for choosing a starting concentration of As(III)=13.3 mM.

Section: 2.3. Analytical Methods

- XPS method is not presented in this section.

- experiments at different times without or with yeast extract are neither mentioned nor described in this section.

  1. Section 3.1.3. XPS analysis of the formed precipitate needs to be modified.

- As it stands, the whole section gives the impression that the authors have had the opportunity to do this additional study, but don't really know why it was done or exactly what information they got that might be useful for the research that is the focus of the manuscript. It seems that XPS analysis prompt you to perform experiments in the presence of yeast extract. Why was that? Also, isn't the previous identification of tooeleite by XRD sufficient to determine its presence? Does this not indicate the presence of arsenic in the form of As(III)?

- The description of the contents of Fig. 4 and the interpretation of the results obtained are inadequate. Neither the caption of Fig. 4 nor the text provides information on how the obtained binding energies relate to As speciation (except for a reference to one of them).

  1. The layout of some chapters probably needs reediting.

Section 3.2. Enhanced mineral stability by yeast extract  provides information relevant to the content of section 3.3.1. Effect of yeast extract on tooeleite transformation.

XRD identification of the precipitate is also a mineralogical characteristic so the title of section 3.3. Mineralogical characterization and As stabilization mechanism seems to be unfortunate.

  1. Section: 3.3. Mineralogical characterization and As stabilization mechanism

Please provide the information what pH and what Fe/As concentrations were used for the experiments at different times.

Please provide information on whether these time experiments were also conducted in the presence of bacteria. Did you run any blanks? How did the duration of the experiment affect the growth of the bacteria in the blank?

  1. line 225

Section:

In the system without yeast extract, the bands at 1195 cm-1, 1127 225 cm-1, and 1042 cm-1 gradually disappeared, and then the band at 1104 cm-1 occurred over 226 time (Fig. 8a), then as shown in Fig. 8c, two bands (1124 cm-1 and 1040 cm-1) were con-227 verted into another band (1100 cm-1), these bands all belong to SO42-.

must be rewritten from scratch with clear language. It is impossible to describe in one sentence all the changes observed in eight spectra showing both changes over time and the effect of the presence of yeast extract. I suggest:

- first to compare the spectrum with literature data to identify its components and confirm that it is a tooeleite spectrum,

- then to determine if there are also other effects on the spectrum indicating other components or impurities etc.

- to describe if there are any changes over time (Fig. a and c)  

In a later text it is also necessary to state why the 3h sample was chosen to show the yeast extract effect before you describe the observed spectral effects of the presence of yeast extract.

The choice of the order of the spectra in Fig. 8 is unfortunate. In the text, spectra 8a and 8c are discussed first, followed by 8b and 8d. This could be simplified.

  1. sections: 3.4. Environmental implications or 4. Conclusion

 - Please refer achieved leaching levels to current norms and environmental legal requirements. The U.S. Environmental Protection Agency (EPA) proposed lowering the maximum contaminant level (MCL) for arsenic from 50 down to 5 micrograms/L (Federal Register, 2000). This followed the recommendation of the National Academy of Sciences report, which concluded that the current 50 ppb standard was not sufficiently protective of public health, and should be lowered. There are also WHO recommendations. Please indicate how relevant are concentrations achieved by this method to these requirements.

- do you have any indications from the literature or previous studies how this I efficient at various As initial concentrations?

Reviewer 2 Report

Overall this manuscript is well-organized and the topic of removing As is attractive. Some minor concerns need to be addressed though:

  1. In the Introduction, data/conclusions from others using Acidithiobacillus ferrooxidans should be added.
  2. In Figure 4, use broader BE range of the precipitate in XPS, for a better understanding of the components. Also, explain the 43.92eV and 44.76eV for different status of As. If possible, add the XPS of precipitate with yeast extract.
  3.  Is the 0.1% yeast extract is the highest? Will a extract percentage higher than 0.1% obtain a better results?
  4. Will the delay of the incorporation by yeast extract affect the production efficiency?
  5. If possible, change the Table 1 to a diagram, for a better overview of the size distribution.
  6. Highlight the amorphous iron in XPS results and description for both using (no amorphous iron in this case?) or not using yeast extraction. Not much difference was shown on FTIR and XRD, which is a concern.

Reviewer 3 Report

This manuscript examined the effect of yeast extract addition on the function of biogenic tooeleite formation by an iron(II)-oxidizing bacterium A. ferrooxidans. The results obtained are interesting and significant for considering As attenuation in AMD environment.

L14. The bacterial name should be written in italics.

L66-68. How do the organic substances contribute to the scorodite formation and As immobilization? Please mention about it. In addition, authors should mention about the similarity or difference between the previous (ref. 29) and present studies in Discussion section. Can such mechanism (if proposed) be applicable to the interpretation of the results obtained here?

L98. What were the model name and supplier of the ICP-OES instrument?

L98. “Dissolved Fe” appears to be “dissolved Fe(II)”. Please include the reference for the titration method.

L114. How were the solid samples collected and then processed before use? Were these analyzed just after collecting?

L115. Please state the compositions of extraction fluid.

L130-131 and Figure 1. “concentrations of dissolved Fe(II)” appears to be appropriate.

L151. Some sentences have disappeared.

L159-, L172-173. Did the various Fe/As conditions affect the bacterial Fe(II) oxidation and Fe precipitation kinetics? This kinetics appear to be important for determining As removal and leaching efficiencies.

L191-. Similarly, the yeast extract addition possibly affects the bacterial Fe oxidation and precipitation, so that it can directly alter the fate of As. I suggest that authors show the data of dissolve Fe(II) concentration during the incubation under various Fe/As conditions and in the presence of yeast extract.

L258-260, 266-267. I cannot understand why yeast extract provides cells with an EPS reaction region. Please state the reason why it is rational. Is it likely that yeast extract stimulates the EPS production by the bacterium or serves as EPS on the cellular surface?

The file of Fig. S1 and Table S1 were not able to be opened.

Round 2

Reviewer 1 Report

Review #2

based on the “Response to Reviewer 1 Comments”

Dear Authors,

Thank you for correctly and accurately responding to the comments and providing extensive explanations. However, the Reviewer expected these clarifications to be in the text of the manuscript and not just in the response to reviewer. This is important information and explanations to allow the reader to understand the logic of the procedure and interpretation. Therefore, please consider the following minor changes to the manuscript based on the information you have already provided in your response (see also a .pdf  file):

Points 1:Section: 2.2. Biotooeleite crystallization experiment

Please give an explanation as to what the rationale was for choosing a starting concentration of As(III)=13.3 mM.

Response 1:We are very grateful for the reviewer’s assessment about our work. The concentration of Arsenic in AMD is in the range of tens to hundreds milligrams per liter, especially, high As enrichments up to 13.3mmol/L (i.e. 1g/L) are also reported from mine drainage sites [1-4]. Therefore, we choose the starting As(III) concentration of 13.3 mM.

Please modify the relevant paragraph accordingly, for example:

The bacterial suspension (10% (v/v)) was inoculated into 300 mL flasks after being filtered by Whatman filter paper to remove precipitates. The 300 mL flasks contained 100 mL of pre-sterilized medium solution containing As(III) (13.3 mM), Fe(II) and yeast extract. An initial As(III) concentration of 13.3 mM was chosen because such high enrichment of up to 1g As(III)/L has been reported at mine drainage sites [1-4]. The mixed solution was stirred at 30℃ on a rotary shaker (170 rpm) for 4 days.

Section:2.3. Analytical Methods

- XPS method is not presented in this section.

Points 2: Section 3.1.3. XPS analysis of the formed precipitate needs to be modified.

- As it stands, the whole section gives the impression that the authors have had the opportunity to do this additional study, but don't really know why it was done or exactly what information they got that might be useful for the research that is the focus of the manuscript. It seems that XPS analysis prompt you to perform experiments in the presence of yeast extract. Why was that? Also, isn't the previous identification of tooeleite by XRD sufficient to determine its presence? Does this not indicate the presence of arsenic in the form of As(III)?

- The description of the contents of Fig. 4 and the interpretation of the results obtained are inadequate. Neither the caption of Fig. 4 nor the text provides information on how the obtained binding energies relate to As speciation (except for a reference to one of them).

Response 2: Thank you for pointing this out. The main purpose of Section 3.1 is to ensure the efficient removal of arsenic before adding yeast extract and highlight the potential of yeast extract for As stabilization, so the optimum condition of biogenic tooeleite need to be determined firstly. This description has been added in Page 9, Line 151-153. As we can see, arsenic removal efficiency achieved the maximum under initial As(III) concentration of 13.3mM, Fe/As=6 and pH of 2.0. XRD patterns show that the precipitates obtained were tooeleite, and it was the only crystalline phase. XPS results further confirm that the valence of As in the precipitates was +3. After confirming its maximum removal efficiency and the formation of biogenic tooeleite, yeast extract was added under Fe/As=6 and pH of 2.0 to verify the mineral stability.

Aimed at the description of the contents of Fig. 4, the description of “The binding energy of As is approximately 44.03 eV. According to the X-ray photoelectron spectroscopy manual [36, 37], As in the precipitate existed as As(III).” has been revised to “An obvious peak is observed with binding energy of 44.03 (eV), thus indicating that the As valence in precipitate is +3 [36, 37]. No signal for As(V) in the XPS was observed.” in Page 11, Line 208-210.

It seems that the Authors still do not understand why XPS analysis shows potential for the direct removal of As(III). I suggest you modify the relevant paragraph indicating why XPS was performed, for example:

The XPS spectrum for the sample obtained at pH 2.0 and Fe/As=6 is shown in Fig. 4. An obvious peak is observed with binding energy of 44.03 (eV), thus indicating that the As valence in precipitate is +3 [37, 38]. No signal for As(V) in the XPS was observed. This confirms the XRD analysis and indicates that the oxidation of As(III) to As(V) does not accompany the removal mechanism. It shows certain potential for the direct removal of As(III) by the formation of biogenic tooeleite. The slightly higher As leaching concentration of tooeleite may substantially restrict its application. Thus the stability of tooeleite was improved by adding yeast extract in the following experiment.

Still the caption for Fig. 4 does not say how the obtained binding energy relates to As speciation. Please modify it, for example:

XPS spectrum showing that the only valence of As in the precipitate obtained under optimal conditions is +3.

Points 4: Section: 3.3. Mineralogical characterization and As stabilization mechanism

Did you run any blanks? How did the duration of the experiment affect the growth of the bacteria in the blank?

Still, there is no information in the manuscript about the use of blanks or how the duration of the experiment affected bacterial growth in blanks. This information is not needed by the reviewer, it is needed by the reader. Please include this graph in supplementary materials and refer properly to the previous work, for example:

Yeast extract had no effect on bacterial growth (Fig. S2) and in experiments conducted in the absence of bacteria, Fe(II) and As(III) concentrations were nearly constant [5].

Figure. S2. Dynamic changes of OD600 under 0.00% and 0.10% yeast extract.
